# Autonomous Capability Assessment of Sequential Decision-Making Systems in Stochastic Settings

**Pulkit Verma, Rushang Karia,** and **Siddharth Srivastava**
Autonomous Agents and Intelligent Robots Lab,
School of Computing and Augmented Intelligence,
Arizona State University, AZ, USA
{verma.pulkit, rushang.karia, siddharths}@asu.edu

## Abstract

It is essential for users to understand what their AI systems can and can't do in order to use them safely. However, the problem of enabling users to assess AI systems with sequential decision-making (SDM) capabilities is relatively understudied. This paper presents a new approach for modeling the capabilities of black-box AI systems that can plan and act, along with the possible effects and requirements for executing those capabilities in stochastic settings. We present an active-learning approach that can effectively interact with a black-box SDM system and learn an interpretable probabilistic model describing its capabilities. Theoretical analysis of the approach identifies the conditions under which the learning process is guaranteed to converge to the correct model of the agent; empirical evaluations on different agents and simulated scenarios show that this approach is few-shot generalizable and can effectively describe the capabilities of arbitrary black-box SDM agents in a sample-efficient manner.

## 1   Introduction

AI systems are becoming increasingly complex, and it is becoming difficult even for AI experts to ascertain the limits and capabilities of such systems, as they often use black-box policies for their decision-making process [Popov et al., 2017, Greydanus et al., 2018]. E.g., consider an elderly couple with a household robot that learns and adapts to their specific household. How would they determine what it can do, what effects their commands would have, and under what conditions? Although we are making steady progress on learning for sequential decision-making (SDM), the problem of enabling users to understand the limits and capabilities of their SDM systems is largely unaddressed. Moreover, as the example above illustrates, the absence of reliable approaches for user-driven capability assessment of AI systems limits their inclusivity and real-world deployability.

This paper presents a new approach for *Query-based Autonomous Capability Estimation* (QACE) of black-box SDM systems in stochastic settings. Our approach uses a restricted form of interaction with the input SDM agent (referred to as SDMA) to learn a probabilistic model of its capabilities. The learned model captures high-level user-interpretable capabilities, such as the conditions under which an autonomous vehicle could back out of a garage, or reach a certain target location, along with the probabilities of possible outcomes of executing each such capability. The resulting learned models directly provide interpretable representations of the scope of SDMA's capabilities. They can also be used to enable and support approaches for explaining SDMA's behavior that require closed-form models (e.g., Sreedharan et al. [2018]). We assume that the input SDMA provides a minimal query-response interface that is already commonly supported by contemporary SDM systems. In particular, SDMA should reveal capability names defining how each of its capabilities

can be invoked, and it should be able to accept user-defined instructions in the form of sequences of such capabilities. These requirements are typically supported by SDM systems by definition.

The main technical problem for QACE is to automatically compute "queries" in the form of instruction sequences and policies, and to learn a probabilistic model for each capability based on SDMA's "responses" in the form of executions. Depending on the scenario, these executions can be in the real world, or in a simulator for safety-critical settings. Since the set of possible queries of this form is exponential in the state space, naïve approaches for enumerating and selecting useful queries based on information gain metrics are infeasible.

**Main contributions**   This paper presents the first approach for query-based assessment of SDMAs in stochastic settings with minimal assumptions on SDMA internals. In addition, it is also the first approach for reducing query synthesis for SDMA assessment to full-observable non-deterministic (FOND) planning [Cimatti et al., 1998]. Empirical evaluation shows that these contributions enable our approach to carry out scalable assessment in both embodied and vanilla SDMAs.

We express the learned models using an input concept vocabulary that is known to the target user group. Such vocabularies span multiple tasks and environments. They can be acquired through parallel streams of research on interactive concept acquisition [Kim et al., 2015, Lage and Doshi-Velez, 2020] or explained to users through demonstrations and training [Schulze et al., 2000]. These concepts can be modeled as binary-valued *predicates* that have their associated evaluation functions [Mao et al., 2022]. We use the syntax and semantics of a well-established relational SDM model representation language, Probabilistic Planning Domain Definition Language (PPDDL) [Younes and Littman, 2004], to express the learned models.

Related work on the problem addresses model learning from passively collected observations of agent behavior [Pasula et al., 2007, Martínez et al., 2016, Juba and Stern, 2022]; and by exploring the state space using simulators [Chitnis et al., 2021, Mao et al., 2022]. However, passive learning approaches can learn incorrect models as they do not have the ability to generate interventional or counterfactual data; exploration techniques can be sample inefficient because they don't take into account uncertainty and incompleteness in the model being learned to guide their exploration (see Sec. 7 for a greater discussion).

In addition to the key contributions mentioned earlier, our results (Sec. 6) show that the approaches for query synthesis in this paper do not place any additional requirements on black-box SDMAs but significantly improve the following factors: (i) convergence rate and sample efficiency for learning relational models of SDMAs with complex capabilities, (ii) few-shot generalizability of learned models to larger environments, and (iii) accuracy of the learned model w.r.t. the ground truth SDMA capabilities. convergence rate to the sound and complete model.

## 2   Preliminaries

**SDMA setup**   We consider SDMAs that operate in stochastic and fully observable environments. An SDMA can be represented as a 3-tuple $\langle \mathcal{X}, \mathcal{C}, \mathcal{T} \rangle$, where $\mathcal{X}$ is the environment state space that the SDMA operates in, $\mathcal{C}$ is the set of SDMA's capabilities (capability names, e.g., "place object x at location y" or "arrange table x") that the SDMA can execute, and $\mathcal{T} : \mathcal{X} \times \mathcal{C} \to \mu\mathcal{X}$ is the stochastic black-box transition model determining the effects of SDMA's capabilities on the environment. Here, $\mu\mathcal{X}$ is the space of probability distributions on $\mathcal{X}$. Note that the semantics of $\mathcal{C}$ are not known to the user(s) and $\mathcal{X}$ may not be user-interpretable. The only information available about the SDMA is the instruction set in the form of capability names, represented as $\mathcal{C}_N$. This isn't a restricting assumption as the SDMAs must reveal their instruction sets for usability.

**Running Example**   Consider a cafe server robot that can pick and place items like plates, cans, etc., from various locations in the cafe, like the counter, tables, etc., and also move between these locations. A capability `pick-item (?location ?item)` would allow a user to instruct the robot to pick up an item like a soda can for any location. However, without knowing its description, the user would not know under what conditions the robot could execute this capability and what the effects will be.

**Object-centric concept representation**   We aim to learn representations that are generalizable, i.e., the transition dynamics learned should be impervious to environment-specific properties such as numbers and configurations of objects. Additionally, the learned capability models should hold in different settings of objects in the environment as long as the SDMA's capabilities does not change. To this

effect, we learn the SDMA's transition model in terms of interpretable concepts that can be represented using first-order logic *predicates*. This is a common formalism for expressing the symbolic models of SDMAs [Zhi-Xuan et al., 2020, Mao et al., 2022]. We formally represent them using a set of object-centric predicates $\mathcal{P}$. The set of predicates used for cafe server robot in Fig. 1 can be `(robot-at ?location)`, `(empty-arm)`, `(has-charge)`, `(at ?location ?item)`, and `(holding ?item)`. Here, `?` precedes an argument that can be replaced by an object in the environment. E.g., `(robot-at tableRed)` means "robot is at the red table." As mentioned

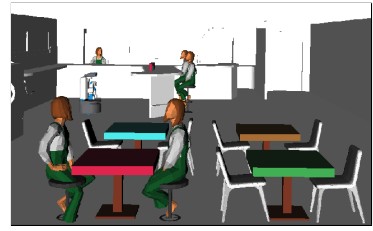

Figure 1: The cafe server robot environment in OpenRave simulator.

earlier, we assume these predicates along with their Boolean evaluation functions (which evaluate to true if predicate is true in a state) are available as input. Learning such predicates is also an interesting but orthogonal direction of research [Mao et al., 2022, Sreedharan et al., 2022, Das et al., 2023].

**Abstraction** Using an object-centric predicate representation induces an abstraction of environment states $\mathcal{X}$ to high-level logical states $\mathcal{S}$ expressible in predicate vocabulary $\mathcal{P}$. This abstraction can be formalized using a surjective function $f : \mathcal{X} \to \mathcal{S}$. E.g., in the cafe server robot, the concrete state $x$ may refer to roll, pitch, and yaw values. On the other hand, the abstract state $s$ corresponding to $x$ will consist of truth values of all the predicates [Srivastava et al., 2014, 2016, Mao et al., 2022].

**Probabilistic transition model** Abstraction induces an abstract transition model $\mathcal{T}' : \mathcal{S} \times \mathcal{C} \to \mu\mathcal{S}$, where $\mu\mathcal{S}$ is the space of probability distributions on $\mathcal{S}$. This is done by converting each transition $\langle x, c, x'\rangle \in \mathcal{T}$ to $\langle s, c, s'\rangle \in \mathcal{T}'$ using predicate evaluators such that $f(x) = s$ and $f(x') = s'$. Now, $\mathcal{T}'$ can be expressed as model $M$ that is a set of parameterized action (capability in our case) schema, where each $c \in \mathcal{C}$ is described as $c = \langle name(c), pre(c), eff(c)\rangle$, where $name(c) \in \mathcal{C}_N$ refers to name and arguments (parameters) of $c$; $pre(c)$ refers to the preconditions of the capability $c$ represented as a conjunctive formula defined over $\mathcal{P}$ that must be true in a state to execute $c$; and $eff(c)$ refers to the set of conjunctive formulas over $\mathcal{P}$, each of which becomes true on executing $c$ with an associated probability. The result of executing

```
(:capability pick-item
 :parameters (?location ?item)
 :precondition (and
   (empty-arm) (has-charge)
   (robot-at ?location)
   (at ?location ?item))
 :effect (and (probabilistic
   0.7 (and (not (empty-arm))
       (not (at ?location ?item))
       (holding ?item))
   0.2 (and (not (has-charge)))
   0.1 (and)))         #No-change
```

Figure 2: PPDDL description for the cafe server robot's *pick-item* capability.

$c$ for a model $M$ is a state $c(s) = s'$ such that $P_M(s'|s, c) > 0$ and one (and only one) of the effects of $c$ becomes true in $s'$. We also use $\langle s, c, s'\rangle$ triplet to refer to $c(s) = s'$. This representation is similar to the Probabilistic Planning Domain Definition Language (PPDDL), which can compactly describe the SDMA's capabilities. E.g., the cafe server robot has three capabilities (shown here as `name(args)`): `pick-item(?location ?item)`; `place-item(?location ?item)`; and `move(?source ?destination)`. The description of `pick-item` in PPDDL is shown in Fig. 2.

**Variational Distance** Given a black-box SDMA $\mathcal{A}$, we learn the probabilistic model $M$ representing its capabilities. To measure how close $M$ is to the true SDMA transition model $\mathcal{T}'$, we use variational distance – a standard measure in probabilistic-model learning literature [Pasula et al., 2007, Martínez et al., 2016, Ng and Petrick, 2019, Chitnis et al., 2021]. It is based on the *total variation distance* between two probability distributions $\mathcal{T}'$ and $M$, given as:

$$\delta(\mathcal{T}', M) = \frac{1}{|\mathcal{D}|} \sum_{\langle s, c, s'\rangle \in \mathcal{D}} \left| P_{\mathcal{T}'}(s'|s, c) - P_M(s'|s, c) \right| \tag{1}$$

where $\mathcal{D}$ is the set of test samples ($\langle s, c, s'\rangle$ triplets) that we generate using $\mathcal{T}'$ to measure the accuracy of our approach. As shown by Pinsker [1964], $\delta(\mathcal{T}', M) \leq \sqrt{0.5 \times D_{KL}(\mathcal{T}' \| M)}$, where $D_{KL}$ is the KL divergence.

## 3 The Capability Assessment Task

In this work, we aim to learn a probabilistic transition model $\mathcal{T}'$ of a black-box SDMA as a model $M$, given a set of user-interpretable concepts as predicates $\mathcal{P}$ along with their evaluation functions, and the capability names $C_N$ corresponding to the SDMA's capabilities. Formally, the assessment task is:

**Definition 1.** *Given a set of predicates $\mathcal{P}$ along with their Boolean evaluation functions, capability names $\mathcal{C}_N$, and a black-box SDMA $\mathcal{A}$ in a fully observable, stochastic, and static environment, the capability assessment task $\langle \mathcal{A}, \mathcal{P}, \mathcal{C}_N, \mathcal{T}' \rangle$ is defined as the task of learning the probabilistic transition model $\mathcal{T}'$ of the SDMA $\mathcal{A}$ expressed using $\mathcal{P}$.*

The solution to this task is a model $M$ that should ideally be the same as $\mathcal{T}'$ for correctness. In practice, $\mathcal{T}'$ need not be in PPDDL, so the correctness should be evaluated along multiple dimensions.

**Notions of model correctness** As discussed in Sec. 2, variational distance is one way to capture the correctness of the learned model. This is useful when the learned model and the SDMA's model are not in the same representation. The correctness of a model can also be measured using qualitative properties such as soundness and completeness. The learned model $M$ should be sound and complete w.r.t. the SDMA's high-level model $\mathcal{T}'$, i.e., for all combinations of $c$, $s$, and $s'$, if a transition $\langle s, c, s' \rangle$ is possible according to $\mathcal{T}'$, then it should also be possible under $M$, and vice versa. Here, $\langle s, c, s' \rangle$ is consistent with $M$ (or $\mathcal{T}'$) if $P(s'|s,c) > 0$ according to $M$ (or $\mathcal{T}'$). We formally define this as:

**Definition 2.** *Let $\langle \mathcal{A}, \mathcal{P}, \mathcal{C}_N, \mathcal{T} \rangle$ be a capability assessment task with a learned model $M$ as its solution. $M$ is* sound *iff each transition $\langle s, c, s' \rangle$ consistent with $M$ is also consistent with $\mathcal{T}'$. $M$ is* complete *iff every transition that is consistent with $\mathcal{T}'$ is also consistent with $M$.*

This also means that if $\mathcal{T}'$ is also a PPDDL model, then (i) any precondition or effect learned as part of $M$ is also present in $\mathcal{T}'$ (soundness), and; (ii) all the preconditions and effects present in $\mathcal{T}'$ should be present in $M$ (completeness). Additionally, a probabilistic model is *correct* if it is sound and complete, and the probabilities for each effect set in each of its capabilities are the same as that of $\mathcal{T}'$.

## 4 Interactive Capability Assessment

To solve the capability assessment task, we must identify the preconditions and effects of each capability in terms of conjunctive formulae expressed over $\mathcal{P}$. At a very high-level, we do this by identifying that a probabilistic model can be expressed as a set of capabilities $c \in C$, each of which has two places where we can add a predicate $p$, namely precondition and effect. We call these *locations* within each capability. We then enumerate through these $2 \times |\mathcal{C}|$ locations and figure out the correct form of each predicate at each of those locations. To do this we need to consider three forms: (i) adding it as $p$, i.e., the predicate must be true for that capability to execute (when the location is precondition), or it becomes true on executing it (when the location is effect); (ii) adding it as *not(p)*, i.e., the predicate must be false for that capability to execute (when the location is precondition), or it becomes false on executing it (when the location is effect); (iii) not adding it at all, i.e., the capability execution does not depend on it (when the location is precondition), or the capability does not modify it (when the location is effect).

**Model pruning** Let $\mathcal{M}$ represent the set of all possible transition models expressible in terms of $\mathcal{P}$ and $\mathcal{C}$. We must prune the set of possible models to solve the capability assessment task, ideally bringing it to a singleton. We achieve this by posing queries to the SDMA and using the responses to the queries as data to eliminate the inconsistent models from the set of possible models $\mathcal{M}$.

Given a location (precondition or effect in a capability), the set of models corresponding to a predicate will consist of 3 transition models: one each corresponding to the three ways we can add the predicate in that location. We call these three possible models $M_T$, $M_F$, $M_I$, corresponding to adding $p$ (true), *not(p)* (false), and not adding $p$ (ignored), respectively at that location.

Note that the actual set of possible transition models is infinite due to the probabilities associated with each transition. To simplify this, we first constrain the set of possible models by ignoring the probabilities, and learn a non-deterministic transition model (commonly referred to as a FOND model [Cimatti et al., 1998]) instead of a probabilistic one. We later learn the probabilities using maximum likelihood estimation based on the transitions observed as part of the query responses.

**Simulator use** Using the standard assumption of a simulator's availability in research on SDM, QACE solves the capability assessment task (Sec. 3) by issuing queries to the SDMA and observing its responses in the form of its execution in the simulator. In non-safety-critical scenarios, this approach can work without a simulator too. The interface required to answer the queries is rudimentary as the SDMA $\mathcal{A}$ need not have access to its transition model $\mathcal{T}'$ (or $\mathcal{T}$). Rather, it should be able to interact with the environment (or a simulator) to answer the queries. We next present the types of queries

we use, followed by algorithms for generating them and for inferring the SDMA's model using its responses to the queries.

**Policy simulation queries ($Q_{PS}$)** These queries ask the SDMA $\mathcal{A}$ to execute a given policy multiple times. More precisely, a $Q_{PS}$ query is a tuple $\langle s_I, \pi, G, \alpha, \eta \rangle$ where $s_I \in \mathcal{S}$ is a state, $\pi$ is a partial policy that maps each reachable state to a capability, $G$ is a logical predicate formula that expresses a stopping condition, $\alpha$ is an execution cutoff bound representing the maximum number of execution steps, and $\eta$ is an attempt limit. Note that the query (including the policy) is created entirely by our solution approach without any interaction with the SDMA. $Q_{PS}$ queries ask $\mathcal{A}$ to execute $\pi$, $\eta$ times. In each iteration, execution continues until either the stopping goal condition $G$ or the execution bound $\alpha$ is reached. E.g., "Given that the robot, `soda-can`, `plate1`, `bowl3` are at `table4`, what will happen if the robot follows the following policy: if there is an item on the table and arm is empty, pick up the item; if an item is in the hand and location is not dishwasher, move to the dishwasher; if an item is in the hand and location is dishwasher, place the item in the dishwasher?" Such queries will be used to learn both preconditions and effects (Sec. 4.3).

A response to such queries is an execution in the simulator and $\eta$ traces of these simulator executions. Formally, the response $\theta_{PS}$ for a query $q_{PS} \in Q_{PS}$ is a tuple $\langle b, \zeta \rangle$, where $b \in \{\top, \bot\}$ indicates weather if the SDMA reached a goal state $s_G \models G$, and $\zeta$ are the corresponding triplets $\langle s, c, s' \rangle$ generated during the $\eta$ policy executions. If the SDMA reaches $s_G$ even once during the $\eta$ simulations, $b$ is $\top$, representing that the goal can be reached using this policy. Next, we discuss how these responses are used to prune the set of possible models and learn the correct transition model of the SDMA.

## 4.1 Query-based Autonomous Capability Estimation (QACE) Algorithm

We now discuss how we solve the capability assessment task using the Query-based Autonomous Capability Estimation algorithm (Alg. 1), which works in two phases. In the first phase, QACE learns all capabilities' preconditions and non-deterministic effects using the policy simulation queries (Sec. 4.2). In the second phase, QACE converts the non-deterministic effects of capabilities into probabilistic effects (Sec. 4.3). We now explain the learning portion (lines 3-11) in detail.

QACE first initializes a model $M^*$ over the predicates in $\mathcal{P}$ with capabilities having names $c_N \in \mathcal{C}_N$. All the preconditions and effects for all capabilities are empty in this initial model. QACE uses $M^*$ to maintain the current partially learned model. QACE iterates over all combinations of $L$ and $\mathcal{P}$ (line 4). For each pair, QACE creates 3 candidate models $M_T$, $M_F$, and $M_I$ as mentioned earlier. It then takes 2 of these (line 5) and generates a query $q$ (line 6) such that responses to the query $q$ from the two models are logically inconsistent (see Sec. 4.2). The query $q$ is then posed to the SDMA $\mathcal{A}$ whose response is stored as $\theta_{\mathcal{A}}$ (line 7). QACE finally prunes at least one of the two models by comparing their responses (which are logically inconsistent) with the response $\theta_{\mathcal{A}}$ of the SDMA on that query (line 8). QACE also updates the effects of all models in the set of possible models to speed up the learning process (line 9). Finally, it learns the probabilities of the observed stochastic effects using maximum likelihood estimation (line 10). An important feature of the algorithm (similar to PLEX [Mehta et al., 2011] and AIA [Verma et al., 2021]) is that it keeps track of all the locations where it hasn't identified the correct way of adding a predicate. The next section presents our approach for generating the queries in line 6.

---

**Algorithm 1:** QACE Algorithm

**Input** : predicates $\mathcal{P}$; capability names $\mathcal{C}_N$; state $s$; SDMA $\mathcal{A}$; hyperparameters $\alpha, \eta$; FOND Planner $\rho$

**Output :** $M$

1   $L \leftarrow \{pre, eff\} \times \mathcal{C}_N$
2   $M^* \leftarrow$ initializeModel $(\mathcal{P}, \mathcal{C}_N)$
3   **for** each $\langle l, p \rangle \in \langle L, \mathcal{P} \rangle$ **do**
4     Generate $M_T, M_F, M_I$ by setting $p$ at $l$ in $M^*$
5     **for** each pair $M_i, M_j$ in $\{M_T, M_F, M_I\}$ **do**
6       $q \leftarrow$ generateQuery$(M_i, M_j, \alpha, \eta, s, \rho)$
7       $\theta_{\mathcal{A}}, \mathbb{S} \leftarrow$ getResponse$(q, \mathcal{A}, s)$
8       $M^* \leftarrow$ pruneModels $(\theta_{\mathcal{A}}, M_i, M_j)$
9       $M^* \leftarrow$ learn possible stochastic effects of capability with $c_N$ in $l$ using $\zeta$ (in $\theta_{\mathcal{A}}$)
10   $M \leftarrow$ learnProbabilitiesOfStochasticEffects$(\zeta, M^*)$
11   **return** $M$

---

## 4.2 Algorithms for Query Synthesis

One of the main challenges in interactive model learning is to generate the queries we discussed above and to learn the agent's model using them. Although active learning [Settles, 2012] addresses the

related problem of figuring out which data sets to request labels for, vanilla active learning approaches are not directly applicable here because the possible set of queries expressible using the literals in a domain is vast: it is the set of all policies expressible using $\mathcal{C}_N$. Query-based learning approaches use an estimate of the utility of a query to select "good" queries. This can be a multi-valued measure like *information gain* [Sollich and Saad, 1994], *value* [Macke et al., 2021], etc. or a binary-valued attribute like *distinguishability* [Verma et al., 2021], etc. We use distinguishability as a measure to identify useful queries. According to it, a query $q$ is distinguishing w.r.t. two models if responses by both models to $q$ are logically inconsistent. We now discuss methods for generating such queries.

**Generating distinguishing queries** QACE automates the generation of queries using search. As part of the algorithm, a model $M^*$ is used to generate the three possible models $M_T$, $M_F$, and $M_I$ for a specific predicate $p$ and location $l$ combination. Other than the predicate $p$ at location $l$, these models are exactly the same. A forward search is used to generate the policy simulation queries with two possible models $M_i, M_j$ chosen randomly from $M_T$, $M_F$, and $M_I$. The forward search is initiated with an initial state $\langle s_0^i, s_0^j \rangle$ as the root of the search tree, where $s_0^i$ and $s_0^j$ are copies of the same state $s_0$ from which we are starting the search. The edges of the tree correspond to the capabilities with arguments replaced with objects in the environment. Nodes correspond to the two states resulting from applying the capability in the parent state according to the two possible models. E.g., consider that a transition $\langle s_0^i, c, s_1^i \rangle$ is possible according to the model $M_i$, and let $\langle s_0^j, c, s_1^j \rangle$ be the corresponding transition (by applying the same effect set of $c$ as $h_i$) according to the model $M_j$. Now there will be an edge in the forward search tree with label $c$ such that parent node is $\langle s_0^i, s_0^j \rangle$ and child node is $\langle s_1^i, s_1^j \rangle$. The search process terminates when a node $\langle s^i, s^j \rangle$ is reached such that either the states $s^i$ and $s^j$ don't match, or the preconditions of the same capability were met in the state according to one of the possible models but not according to the other. Vanilla forward search scales poorly with the number of capabilities and objects in the environment. To address this we reduce the problem to a fully observable non deterministic (FOND) planning problem. This can be solved by any FOND planner. The output of this search is a policy $\pi$ to reach a state where the two models, $M_i$ and $M_j$ predict different outcomes. Additional details about the reduction and an example of the output policy are available in the extended version of the paper [Verma et al., 2023]. The query $\langle s_I, \pi, G, \alpha, \eta \rangle$ resulting from this search is such that $s_I$ is set to the initial state $s_0$, $\pi$ is the output policy, $G$ is the goal state where the models' responses doesn't match, $\alpha$ and $\eta$ are hyperparameters as mentioned earlier. We next see how to use these queries to prune out the incorrect models.

## 4.3 Learning Probabilistic Models Using Query Responses

At this point, QACE already has a query such that the response to the query by the two possible models does not match. We next see how to prune out the model inconsistent with the SDMA. QACE poses the query generated earlier to the SDMA and gets its response (line 7 in Alg. 1). If the SDMA can successfully execute the policy, QACE compares the response of the two models with that of the SDMA and prunes out the model whose response does not match with that of the SDMA. If the SDMA cannot execute the policy, i.e., SDMA fails to execute some capability in the policy, then the models cannot be pruned directly. In such a case, a new initial state $s_0$ must be chosen to generate a new query starting from that initial state. Since generating new queries for the same pair of models can be time consuming, we preempt this issue by creating a pool of states $\mathbb{S}$ that can execute the capabilities using a directed exploration of the state space with the current partially learned model as discussed below.

**Directed exploration** A partially learned model is a model where one or more capabilities have been learned (the correct preconditions have been identified for each capability and at least one effect is learned). Once we have such a model, we can do a directed exploration of the state space for these capabilities by only executing a learned capability if the preconditions are satisfied. This helps in reducing the sample complexity since the simulator is only called when we know that the capability will execute successfully, thereby allowing us to explore different parts of the state space efficiently. If a capability's preconditions are not learned, all of its groundings might need to be executed from the state. In the worst case, to escape local minima where no models can be pruned, we would need to perform a randomized search for a state where a capability is executable by the SDMA. In practice, we observed that using directed exploration to generate a pool of states gives at least one grounded capability instance. This helps ensure that during query generation, the approach does not spend a long time searching for a state where a capability is executable.

**Learning probabilities of stochastic effects** After QACE learns the non-deterministic model, to learn the probabilities of the learned effects it uses the transitions collected as part of responses to queries. This is done using Maximum Likelihood Estimation (MLE) [Fisher, 1922]. For each triplet $\langle s, c, s' \rangle$ seen in the collected data, let $count_c$ be the number of times a capability $c$ is observed. Now, for each effect set, the probability of that effect set becoming true on executing that capability $c$ is given as the number of times that effect is observed on executing $c$ divided by $count_c$. As we increase the value of the hyperparameter $\eta$, we increase the number of collected triplets, thereby improving the probability values calculated using this approach.

## 5 Theoretical Analysis and Correctness

We now discuss how the model $M$ of SDMA $\mathcal{A}$ learned using QACE fulfills the notions of correctness (Sec. 3) discussed earlier. We first show that the model $M^*$ learned before line 10 of QACE (Alg. 1) is sound and complete according to Def. 2. The proofs for the theorems are available in the extended version of the paper [Verma et al., 2023].

**Theorem 1.** *Let $\mathcal{A}$ be a black-box SDMA with a ground truth transition model $\mathcal{T}'$ expressible in terms of predicates $\mathcal{P}$ and a set of capabilities $\mathcal{C}$. Let $M^*$ be the non-deterministic model expressed in terms of predicates $\mathcal{P}^*$ and capabilities $\mathcal{C}$, and learned using the query-based autonomous capability estimation algorithm (Alg. 1) just before line 10. Let $C_N$ be a set of capability names corresponding to capabilities $\mathcal{C}$. If $\mathcal{P}^* \subseteq \mathcal{P}$, then the model $M^*$ is* sound *w.r.t. the SDMA transition model $\mathcal{T}'$. Additionally, if $\mathcal{P}^* = \mathcal{P}$, then the model $M^*$ is* complete *w.r.t. the SDMA transition model $\mathcal{T}'$.*

Next, we show that the final step of learning the probabilities for all the effects in each capability converges to the correct probability distribution under the assumption that all the effects of a capability are identifiable. When a capability $c$ is executed in the environment, one of its effects $e_i(c) \in eff(c)$ will be observed in the environment. To learn the correct probability distribution in $M$, we should accurately identify that effect $e_i(c)$. Hence, the set of effects is *identifiable* if at least one state exists in the environment from which each effect can be uniquely identified when the capability is executed. An example of this is available in the extended version of the paper [Verma et al., 2023].

**Theorem 2.** *Let $\mathcal{A}$ be a black-box SDMA with a ground truth transition model $\mathcal{T}'$ expressible in terms of predicates $\mathcal{P}$ and a set of capabilities $\mathcal{C}$. Let $M$ be the probabilistic model expressed in terms of predicates $\mathcal{P}^*$ and capabilities $\mathcal{C}$, and learned using the query-based autonomous capability estimation algorithm (Alg. 1). Let $\mathcal{P} = \mathcal{P}^*$ and $M$ be generated using a sound and complete non-deterministic model $M^*$ in line 11 of Alg. 1, and let all effects of each capability $c \in \mathcal{C}$ be identifiable. The model $M$ is* correct *w.r.t. the model $\mathcal{T}'$ in the limit as $\eta$ tends to $\infty$, where $\eta$ is hyperparameter in query $Q_{\text{PS}}$ used in Alg. 1.*

## 6 Empirical Evaluation

We implemented Alg. 1 in Python to evaluate our approach empirically.[1] We found that our query synthesis and interactive learning process leads to (i) few shot generalization; (ii) convergence to a sound and complete model; and (iii) much greater sample efficiency and accuracy for learning lifted SDM models with complex capabilities as compared to the baseline.

**Setup** We used a *single* training problem with few objects ($\leq 7$) for all methods in our evaluation and used a test set that was composed of problems containing object counts larger than those in the training set. We ran the experiments on a cluster of Intel Xeon E5-2680 v4 CPUs with CentOS 7.9 running at 2.4 GHz with a memory limit of 8 GB and a time limit of 4 hours. We used PRP [Muise et al., 2012] as the FOND planner to generate the queries (line 6 in Alg. 1). For QACE, we used $\alpha = 2d$ where $d$ is the maximum depth of policies used in queries generated by QACE and $\eta = 5$. All of the methods in our empirical evaluation receive the same training and test sets and are evaluated on the same platform. We used Variational Distance (VD) as presented in Eq. 1 to evaluate the quality of the learned SDMA models.

**Baseline selection** We used the closest SOTA related work, GLIB [Chitnis et al., 2021] as a baseline. GLIB learns a probabilistic model of an intrinsically motivated agent by sampling goals far away from the initial state and making the agent try to reach them. This can be adapted to an assessment setting

---

[1]Source code available at `https://github.com/AAIR-lab/QACE`

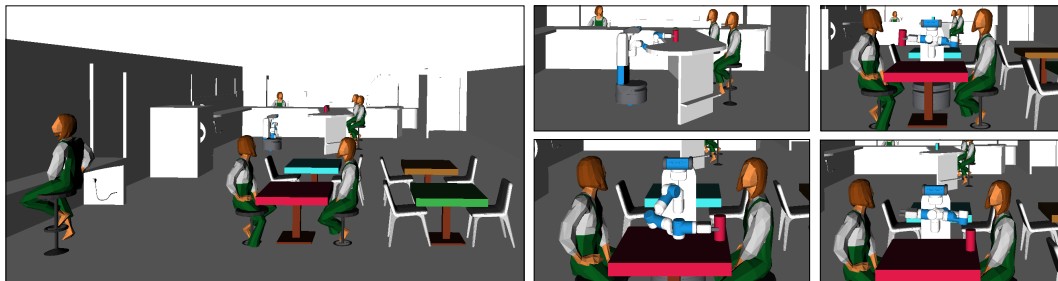

Figure 3: Screen captures from the Cafe Server Robot simulation. The complete environment is shown in the image on the left. The image grid on the right shows screen captures of multiple steps of the robot delivering a `soda-can` to a table.

by moving goal-generation based sampling outside the agent, and, to the best of our knowledge, no existing approach addresses the problem of creating intelligent questions for an SDMA. GLIB has two versions, GLIB-G, which learns the model as a set of grounded noisy deictic rules (NDRs) [Pasula et al., 2007], and GLIB-L, which learns the model as a set of lifted NDRs. We used the same hyperparameters as published for the *Warehouse Robot* and *Driving Agent* and performed extensive tuning for the others and report results with the best performing settings.

The models learned using GLIB cannot be used to calculate the variational distance presented in Eq. 1 because for each capability GLIB learns a set of NDRs rather than a unique NDR. In order to maintain parity in comparison, we use GLIB's setup to calculate an approximation of the VD. Using it, we sample 3500 random transitions $\langle s, c, s' \rangle$ from the ground truth transition model $\mathcal{T}'$ using problems in the test set to compute a dataset of transitions $\mathcal{D}$. The sample-based, approximate VD is then given as: $\frac{1}{|\mathcal{D}|} \sum_{d \in \mathcal{D}} \mathbb{1}_{[s' \neq c_M(s)]}$, where $c_M(s)$ samples the transition using the capability in the learned model output by each method. In Fig. 5, we compare the approximate variational distance of the three approaches w.r.t. $\mathcal{D}$ as we increase the learning time. Note that we also evaluated VD for QACE using Eq. 1 and found that $\delta(\mathcal{T}', M) \approx 0$ for our learned model $M$ in all SDMA settings.

**SDMAs for evaluation** To test the efficacy of our approach, we created SDMAs for five different settings including one task and motion planning agent and several SDMAs based on state-of-the-art stochastic planning systems from the literature: *Cafe Server Robot* is a Fetch robot [Wise et al., 2016] that uses the ATM-MDP task and motion planning system [Shah et al., 2020] to plan and act in a restaurant environment to serve food, clear tables, etc.; *Warehouse Robot* is a robot that can stack, unstack, and manage the boxes in a warehouse; a *Driving Agent* that can drive between locations and can repair the vehicle at certain locations; a *First Responder Robot* that can assist in emergency scenarios by driving to emergency spots, providing first-aid and water to victims, etc.; and an *Elevator Control Agent* that can control the operation of multiple elevators in a building.

Additional details about each setting are available in the extended version [Verma et al., 2023].

## 6.1 Results

We present an analysis of our approach on all of the SDMAs listed above. We also present a comparative analysis with the baseline on all SDMAs except the Cafe Server Robot, whose task and motion planning system was not compatible with the baseline.

**Cafe Server Robot** This SDMA setup uses an 8 degrees of freedom Fetch [Wise et al., 2016] robot in a cafe setting on OpenRave simulator [Diankov and Kuffner, 2008]. The low-level environment state consists of continuous x, y, z, roll, pitch, and yaw values of all objects in the environment. The predicate evaluators were provided by ATM-MDP of which we used only a subset to learn a PPDDL model. Each robot capability is refined into motion controls at run-time depending on the configuration of the objects in the environment. The results for variational distance between the learned model and the ground truth model in Fig. 4 show that despite the different vocabulary, QACE learns an accurate transition model for the SDMA.

We now discuss the comparative performance of QACE with the baseline across the four baseline-compatible SDMAs presented above.

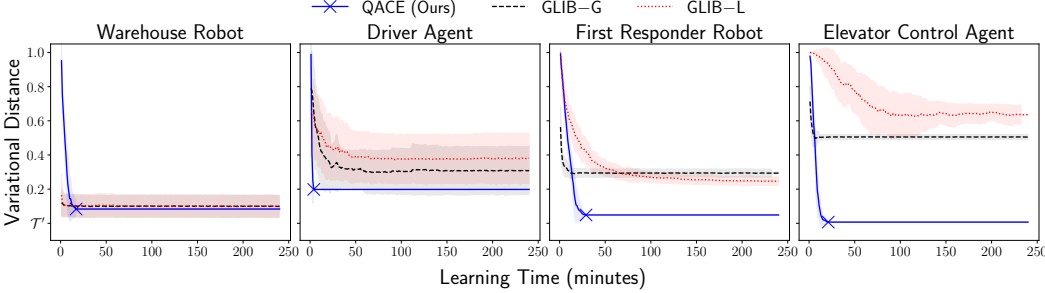

Figure 5: A comparison of the approximate variational distance as a factor of the learning time for the three methods: QACE (ours), GLIB-G, and GLIB-L (lower values better). × shows that the learning process ended at that time instance for QACE. The results were calculated using 30 runs per method per domain. Solid lines are averages across runs, and shaded portions show the standard deviation. $\mathcal{T}'$ is the ground truth model. Detailed results are available in Verma et al. [2023].

**Faster convergence** The time taken for QACE to learn the final model is much lower than that of GLIB for three of the four SDMAs. This is because trace collection by QACE is more directed and hence ends up learning the correct model in *a shorter time*. The only setup where GLIB marginally outperforms QACE is Warehouse Robot, and this happens because this SDMA has just two capabilities, one of which is deterministic. Hence, GLIB can easily learn their configuration from a few observed traces. For SDMAs with complex and much larger number of capabilities – First Responder Robot and Elevator Control Agent – GLIB finds it more challenging to learn the model that is closer to the ground truth transition model. Additionally, QACE takes much fewer samples to learn the model than the baselines. In all settings, QACE is much more *sample efficient* than the baselines as QACE needed at most 4% of the samples needed by GLIB-G to reach the variational distance that it plateaued at. In contrast, GLIB-L started timing out only after processing a few samples for complex SDMAs.

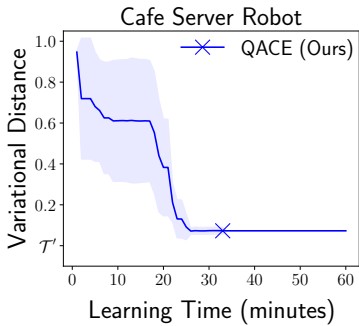

Figure 4: Variational Distance between the learned model and the ground truth with increasing time for QACE for Cafe Server Robot. × shows that the learning process ended at that time instance.

**Few-shot generalization** To ensure that learned models are not overfitted, our test set contains problems with larger quantities of objects than those used during training. As seen in Fig. 5, the baselines have higher variational distance from the ground truth model for complex SDMA setups as compared to QACE. This shows *better few-shot generalization* of QACE compared to the baselines.

## 7 Related Work

The problem of learning probabilistic relational agent models from a given set of observations has been well studied [Pasula et al., 2007, Mourão et al., 2012, Martínez et al., 2016, Juba and Stern, 2022]. Jiménez et al. [2012] and Arora et al. [2018] present comprehensive reviews of such approaches. We next discuss the closest related research directions.

**Passive learning** Several methods learn a probabilistic model of the agent and environment from a given set of agent executions. Pasula et al. [2007] learn the models in the form of noisy deictic rules (NDRs) where an action can correspond to multiple NDRs and also model noise. Mourão et al. [2012] learn such operators using action classifiers to predict the effects of an action. Rodrigues et al. [2011] learn non-deterministic models as a collection of rule sets and learn these rule sets incrementally. They take a bound on the number of rules as input. Juba and Stern [2022] provide a theoretical framework to learn safe probabilistic models with a range of probabilities for each probabilistic effect while assuming that each effect is atomic and independent of others. A common issue with such approaches is that they are susceptible to incorrect and sometimes inefficient model

learning as they cannot control the input data used for learning or carry out interventions required for accurate learning.

**Sampling of transitions**   Several approaches learn operator descriptions by exploring the state space in the restricted setting of deterministic models [Ng and Petrick, 2019, Jin et al., 2022]. A few reinforcement learning approaches have been developed for learning the relational probabilistic action model by exploring the state space using pre-determined criteria to generate better samples [Ng and Petrick, 2019]. Konidaris et al. [2018] explore learning PPDDL models for planning, but they aim to learn the high-level symbols needed to describe a set of input low-level options, and these symbols are not interpretable. GLIB [Chitnis et al., 2021] also learns probabilistic relational models using goal sampling as a heuristic for generating relevant data, whereas we use active querying using guided forward search for this. Our empirical analysis shows that our approach of synthesising queries yield greater sample efficiency and correctness profiles than the goal generation used in this approach.

**Active learning**   Several active learning approaches learn automata representing a system's model [Angluin, 1988, Aarts et al., 2012, Pacharoen et al., 2013, Vaandrager, 2017]. These approaches assume access to a teacher (or an oracle) that can determine whether the learned automaton is correct and provide a counterexample if it is incorrect. This is not possible in the black-box SDMA settings that constitute the focus of this work.

# 8   Conclusion

In this work, we presented an approach for learning a probabilistic model of an agent using interactive querying. We showed that the approach is few-shot generalizable to larger environments and learns a sound and complete model faster than state-of-the-art approaches in a sample-efficient manner.

QACE describes the capabilities of the robot in terms of predicates that the user understands (this includes novice users as well as more advanced users like engineers). Understanding the limits of the capabilities of the robot can help with the safe usage of the robot, and allow better utilization of the capabilities of the robot. Indirectly, this can reduce costs since the robot manufacturer need not consider all possible environments that the robot may possibly operate in. The use of our system can also be extended to formal verification of SDMAs.

QACE can also be used by standard explanation generators as they need an agent's model. Such models are hard to obtain (as we also illustrate in this paper) and our approach can be used to compile them when they are not available to start with.

**Limitations and Future Work**   In this work, we assume that the agent can be connected to a simulator to answer the queries. In some real-world settings, this assumption may be limiting as users might not have direct access to such a simulator. Formalizing the conditions under which is it safe to ask the queries directly to the agent in the real-world is a promising direction for the future work. Additionally, in this work, we assume the availability of the instruction set of the SDMA as input in the form of capability names. In certain settings, it might be useful to discover the capabilities of an evolving SDMA using methods proposed by Nayyar et al. [2022] and Verma et al. [2022].

## Acknowledgments and Disclosure of Funding

We thank Jayesh Nagpal for his help with setting up the Cafe Server Robot SDMA. We also thank anonymous reviewers for their valuable feedback and suggestions. This work was supported by the ONR under grants N00014-21-1-2045 and N00014-23-1-2416.

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
