# OpenReview forum: "Autonomous Capability Assessment of Sequential Decision-Making Systems in Stochastic Settings"
_NeurIPS.cc/2023/Conference — NeurIPS 2023 poster_

### Official Review · Reviewer_k56H · 2023-07-04

**Soundness:** 3 good
**Presentation:** 3 good
**Contribution:** 2 fair
**Rating:** 5
**Confidence:** 2

**Summary:**

This paper introduces QACE, an algorithm for automatically learning the capabilities of sequential decision making agents through distinguishing queries. The problem is formulated in terms of the predicates and capabilities of an agent, which are assumed to be known a priori. QACE then aims to compute a transition model that encodes the probabilities that executing a given capability in a given state s will lead to state s'.

**Strengths:**

**Originality:**
This is an original piece of work (novel combination of techniques) and related work seems to be adequately cited.

**Quality:**
The submission seems to be technically sound and the claims are supported. The authors discuss limitations of their work. The methods are appropriate but it could have been interesting to see a comparison against simpler baselines, such as against an approach that randomly generates queries.


**Clarity:**
The submission is well written. The process for generating distinguishing queries could be demonstrated through figures (added to the supplemental material, for example) that show how the tree is built and how the pruning is executed. Similar strategies could be used to provide examples of the non-deterministic model and how it is transformed into a probabilistic model. More detail on background could be provided, for example with sections (e.g., in the supplemental material) introducing FOND planning and PPDDL. Also, sometimes it feels like some claims could use a bit more explanation (e.g., question c), below).


**Significance:**
I think the results are important, however it is not clear to me who would leverage QACE or how. Is it the actual users? Is it robot designers/engineers?


EDIT:

I read the author's rebuttal, which helped clarify matters I had not fully understood.

**Weaknesses:**

Please see above for strengths and weaknesses.

**Questions:**

**a)**
In the introduction, the authors claim that part of what makes this problem relevant is to allow users to understand what their robots can do and under what conditions. However it is unclear to me how QACE could help a user learn to take the most out of its robot. If instead QACE is meant to be used by a different stakeholder group, then I think this is not clear in the paper.

**b)**
Is a learning time of 20 or 30 minutes a big overhead for a user? Would they feel frustrated and give up from trying to learn how to use the robot? It is difficult to evaluate if the learning time of QACE is acceptable if there is no baseline against which to compare it.

**c)**
[lines 283-284] "we preempt this issue by creating a pool of states S that can execute the capabilities using a directed exploration of the state space using partially learned models.", it is not clear to me how having this pool of states can prevent the generation of queries to take "forever" (in cases where the hypotheses cannot be pruned directly). Also, what is a partially learned model in such cases and how do you get them and what do you mean with "directed exploration of the state space"?

**d)**
[Proof of proposition 1] Why is it not possible, according to Alg.1, for the two models to "either have different preconditions for c′ or different effects." Isn't this necessary for the FOND planning problem to have a solution? "the FOND planning problem ⟨M_ij , s_I_ij ,G_ij⟩, which has a solution if both the models have different precondition or at least one different effect for the same capability."


**Comments:**

An introduction to the basics of FOND planning is missing. Providing one in the supplemental material could be helpful.

Similarly, some background on PPDDL would help with Figure 2 and section 1.3 of the supplemental material.

Figure 5. Standard deviation of QACE is hard to see. For example changing colours could improve readability.

[minor details]
- [line 142] "model M that should ideally be same as T'" ==> **the** same as T'?
- [line 160] "formalae"

**Limitations:**

The authors discuss some limitations of their work.

---

> ### Author Rebuttal · Authors · 2023-08-10
>
> We thank the reviewer for detailed review and suggestions. We plan to use the additional page to incorporate them including suggestions for the description of FOND and PPDDL models.
>
> **Quality:** GLIB uses such a method that generates random traces and hence is used in our comparison as a baseline. We will clarify this in addition to GLIB's summary in the final version.
>
> We would like to highlight that, to the best of our knowledge, no other existing approach is able to solve the problem we are addressing in this work. There are significant challenges in solving the assessment problem we address here due to which the closest available baseline GLIB has several limitations, and we provided the baseline with additional details to work for the setting we address. GLIB requires input goals that should satisfy certain conditions (should be conjunctions of at least three predicates that are not true in the initial state), so we manually provided it with such goals (for each of the 3-5 input problems per domain) for it to perform comparably. We also used the same set of hyperparameters that the GLIB authors used in their evaluation. Even with these changes, we would need the fixes mentioned in lines 352-356 for it to apply to our setting. We will clarify this.
>
> **a)** (and **Significance**) As the reviewer correctly points out, we mentioned in lines 18-19 that in terms of an SDMA, we envision that lay users should be able to determine what an SDMA “can do, what effects their commands would have, and under what conditions?”. In this context, our approach (QACE) describes the capabilities of the robot in terms of predicates that the user understands (This includes novice users as well as more advanced users like engineers.) Understanding the limits of the capabilities of the robot can help with the safe usage of the robot, and allow better utilization of the capabilities of the robot. Indirectly, this can reduce costs since the robot manufacturer need not consider all possible environments that the robot may possibly operate in. The use of our system can also be extended for formal verification of SDMAs.
>
> **b)** We think that a learning time of 20-30 minutes might seem excessive *iff* there is no output provided to the user by the system during the assessment process. Our system works by identifying the correct preconditions and effects, one <l,p> pair at a time. This could be streamed to the user and the user can immediately benefit from this information (which is guaranteed to be correct). Furthermore, assessment is an infrequent process and the system could be programmed to run assessment during large periods of inactivity, thus making sure that the user is always informed of any changes that could occur (e.g. coefficient of friction of wheels changing from wear-and-tear). As seen by our empirical evaluation, our approach can provide the user with a complete model significantly faster than other SOTA approaches that learn models.
>
> You are correct that a baseline performing exactly the same work as ours is missing, and hence we believe this is a valuable contribution in the direction of the personalized assessment of SDMA systems.
>
> Note that our approach can be used by standard explanation generators as they need an agent’s model. Existing methods for explanation generation [1,2] require such models as input. Those models are hard to obtain (as we also illustrate in this paper) and this approach generates those models when they are not available to the users to start with.
>
> **c)** As mentioned in lines 279-284, an important step in pruning the hypotheses is to get a state where the SDMA can execute a capability. Generating such states is easier if we use directed exploration, which can increase the probability of encountering such a state. Once we have such a state, we can use a process similar to the one used in response of Q1 to reviewer *Jehr* above.
>
> *Directed Exploration:* A partially learned model is a model where one or more capabilities have been learned (the correct preconditions have been identified for each capability and at least one effect is learned). We will clarify this. Once we have such a model, we can do a directed exploration of the state space for these capabilities by only executing a learned capability if the preconditions are satisfied. This helps in reducing the sample complexity since the simulator is only called when we know that the capability will execute successfully, thereby allowing us to explore different parts of the state space efficiently. Naturally, if a capability's preconditions are not learned, all of its groundings might need to be executed from the state.
>
> In the worst case, to escape local minimas where no hypotheses can be pruned, we would need to perform a randomized search for a state where a capability is executable by the SDMA. But practically, as we observed in our empirical evaluation, using directed exploration to generate a pool of states gives at least one grounded capability instance. This ensures that during the query generation, the approach need not spend a long time searching for a state where a capability is executable.
>
> **d)** It is not possible for the two models to "either have different preconditions for $c′$ or different effects” because the location $l$ corresponds to capability $c$ in line 124. According to Alg. 1, the models $M_i$ and $M_j$ are created such that they differ in precondition or effect of $c$ (depending on $l$).  Now since $c \neq c’$ and the models already differ in precondition (or effect) of $c$, and the models $M_i$ and $M_j$ are exactly the same other than this difference. Hence $c’$ cannot have different preconditions or effects.
>
> -----
> *References:*
>
> [1] Chakraborti et al. Plan Explanations as Model Reconciliation: Moving Beyond Explanation as Soliloquy. IJCAI 2017.
>
> [2] Eifler et al. Plan-Space Explanation via Plan-Property Dependencies: Faster Algorithms & More Powerful Properties. IJCAI 2020.

---

> > ### Comment · Reviewer_k56H · 2023-08-15
> >
> > Thank you for answering the questions.
> >
> > I confirm that I have read the rebuttal.
> >
> > I believe QACE may contribute to advancing the field and help model capabilities of black-box AI systems.
> >
> > However, since I'm not an expert in the area, and I'm not familiar with FOND and PPDDL models. and current efforts in this domain, I cannot fully appreciate the technical details of the paper nor its insights and as such do not feel comfortable raising my score to a weak accept.

---

> > > ### Author Response · Authors · 2023-08-21
> > >
> > > Thank You for your review. We are glad that our response helped to answer your questions.

---

### Official Review · Reviewer_Gtv9 · 2023-07-05

**Soundness:** 3 good
**Presentation:** 3 good
**Contribution:** 2 fair
**Rating:** 6
**Confidence:** 1

**Summary:**

The paper proposes an algorithm for learning a probabilistic model of a black box agent's capabilities. The method assumes the existence of a vocabulary to describe the environment's state and the set of capabilities. The proposed method generates all possible hypotheses using three ways to add a predicate (as a condition, a negated condition, and not adding it). In the first instance, these hypotheses have no probabilities assigned. Using a sequence of distinguishing queries generated with a planner, they prune the version space (the possibly correct hypotheses). Lastly, the algorithm employs a frequentist estimation of the probabilities associated with the transitions.

The authors validate the algorithm empirically comparing it to a previous SOTA method.

**Strengths:**

The strength of the paper is the careful exposition of an intuitive approach to the problem, along with the empirical evaluation claiming to surpass previous state-of-the-art algorithms.

**Weaknesses:**

The weakness of the paper is the assumption that all possible hypotheses need to be generated. In real-life scenarios, this might be prohibitive.

**Questions:**

The abstract mentions "evolving sequential decision making", but the methods assume the capabilities of the black box are fixed, and it doesn't discuss an evolving/adapting agent.

What is a FOND model? A short description would help.

**Limitations:**

Yes, the authors discussed limitations.

---

> ### Author Rebuttal · Authors · 2023-08-10
>
>  We thank the reviewer for the review and suggestions. We address your concerns below:
>
> **Weakness)** As mentioned in lines 177-181, there are just three hypotheses corresponding to any $\langle l,p \rangle$ pair. Hence the number of hypotheses to be considered are a small constant (=3) at any step in the algorithm.
>
> ----------
> The answer to the individual questions are added below:
>
> **Q1) Evolving SDM:** We wanted to focus that in case an SDM’s capabilities evolve; we would want to generate the descriptions of the SDM agent after the update. The model itself remains the same during the assessment process. We will clarify this point in the paper.
> An example scenario of such evolving capabilities would be a robot that is initially deployed and assessed by QACE to yield a model $M_1$. After some time, the coefficient of friction of the wheels and gripper could change due to wear and tear, changing the model. The assessment process could be re-run (and in fact can be run using overnight batch updates since it is automatic, and handsfree requiring no human intervention) to capture this update to the model.
>
> **Q2) FOND Model:** A FOND model is a fully observable non-deterministic model. Each capability has a precondition similar to the probabilistic model (lines 105-126), and an effect in the FOND model is also similar to the probabilistic model but without the associated probabilities. The capability shown in Figure 2, will be expressed as follows in a FOND model:
> ```
> (:capability pick-item
>   :parameters (?location ?item)
>   :precondition (and (empty-arm)
>     (has-charge)
>     (robot-at ?location)
>     (at ?location ?item)
>   )
> :effect (oneof
>     (and (not (empty-arm)) (not (at ?location ?item)) (holding ?item))
>     (and (not (has-charge)))
>     (and)  # No-change
>   )
> )
> ```
> Here ```oneof``` in the effect represents that only one of the three effects will be applied on executing this capability.

---

> > ### Comment · Reviewer_Gtv9 · 2023-08-20
> > **Thank you for the clarifications**
> >
> > I thank the authors for their clarifications. Without being familiar with the field, I consider the current work solid enough to be published, but since I can't argue more for its impact in its field, I will keep my 'weak accept' suggestion.

---

> > > ### Author Response · Authors · 2023-08-21
> > >
> > > Thank you for your review. We are happy that our response helped in answering your queries.

---

### Official Review · Reviewer_Sjai · 2023-07-07

**Soundness:** 3 good
**Presentation:** 2 fair
**Contribution:** 3 good
**Rating:** 5
**Confidence:** 4

**Summary:**

The paper tackles the problem of modeling the capabilities of a block-box sequential decision-making agent (SDMA) by querying the SDMA agent along the way. The presented method (QACE) uses an active learning approach to interact with the block-box SDMA and learn an interpretable probabilistic model of its capabilities. The paper also presents a theoretical analysis of QACE showing that it can learn a model that is both complete and sound w.r.t the ground-truth model of the SDMA. QACE works using version space partitioning using the queries to remove inconsistent hypotheses until it converges on one model. In the final step, the learned non-deterministic model is converted into probabilities over the capabilities of the system -- via MLE done on the collected data via responses to the queries. QACE is evaluated in four different settings -- Cafe Server robot, warehouse robot, driving agent, and first responder agent. Results show that QACE is able to recover the underlying model of each environment almost always in a reasonable time. The method also outperforms extensions of SOTA methods such as GLIB-L and GLIB-A.

**Strengths:**

The premise of the problem that the paper tackles is an interesting one. It is a very practically relevant problem as the framework outputs interpretable capability names of the SDMA and defines how each capability name can be invoked. The paper is written clearly and the running example of the Cafe-server robot makes it easier to understand the presented algorithm. The experimental setup also shows that the policy simulation queries $\eta (= 5)$ to learn distinguishing queries, are within a reasonable range, to be applied in practical environments. The completeness and soundness guarantees provide a stronger basis for the adoption of the algorithm. Overall, the problem setup is quite interesting, and QACE provides a new direction to reason about the capabilities of block-box SDMA in an interpretable manner.

**Weaknesses:**

- On the scalability of QACE: From the paper, it was not clear as to whether QACE can scale to systems that have a larger space of predicate and therefore larger probabilistic problem domain description language. For systems with larger predicates, would the parameter $\eta$ be reasonable enough to estimate the response to a query?

- Extended Theoretical analysis: While completeness and soundness guarantees are provided, it would also be interesting to see how many iterations of QACE are required for it to provide a reasonable estimate of the model. What is the convergence rate of QACE w.r.t the error in estimating the model?

- Clarity: A short description of the GLIB (the baseline) method would be good, as it is the only baseline that QACE is compared against.

**Questions:**

- Clarification: Is it correct to assume that QACE assumes that there is a unique model that defines a given SDMA? What happens if that assumption is not satisfied, or is it the case that SDMA will always have a unique model by design?

- What happens if the SDMA systems have some margin of error in providing responses to the queries? How would the error propagation be handled when generating distinguishing queries?

- In real systems, sometimes it is not feasible to ask several queries to a system. When the number of queries has a budget (say c number of queries/ or the cost of a query is c), how does it affect the accuracy of the learned model? Can further guarantees be provided on the learned model?

- See "Weaknesses" for additional clarifications.

**Limitations:**

The authors have addressed the limitations of the presented method and in its current form, there is no potential negative societal impact of the work.

---

> ### Author Rebuttal · Authors · 2023-08-10
>
> Thank you for the detailed review. We address your questions and concerns below:
>
> **Q1)** No, QACE doesn’t assume that there is a single model that defines the given SDMA’s functionality. QACE can return a functionally equivalent model when there are multiple correct representations. E.g., if p(x) and q(x) are equisatisfiable in a domain, functionally equivalent models can be created by replacing p's and q's with each other. In such cases, QACE will return a model that is functionally equivalent (modulo such substitutions).
>
> **Q2)** In the current setup, the SDMA responds with the policy or the capability sequence that it would execute in response to the objective in a given query. However, the system allows execution errors and stochastic environments. The system can model probabilistic effects, and as a result execution errors will be learned as additional effects for the respective capability in the query. Thank you for the question. We will clarify this in the final version.
>
> **Q3)** (and **Weakness 2**): For our approach, QACE, the number of iterations of the algorithm is bounded by the number of interactions (steps) the system has with the agent. This is because each iteration has at least one interaction with the agent. The number of interactions (steps) is bounded by $\eta \times \alpha \times num-queries$. The plots for variational distance vs. the number of steps are available in the supplementary material (Fig. 3). Since this number is very small for QACE when compared to the baseline GLIB, we have also included a zoomed-in version of the plots and  a plot of variational distance vs number of unique queries in the additional supplementary material (Sec. 5) submitted with this rebuttal response. We will merge this with the main supplementary in the final version.
>
> Theoretically, for a fixed query budget, we can get an upper and lower bound on the number of $\langle l,p \rangle$ tuples that can be learned correctly. This is because if we stop the learning process in between, the current model M* will be correct in terms of $\langle l,p \rangle$ tuples that were already processed by QACE up to that point.
>
> **Weakness 1)** Scalability Issue: We are providing a table below with the size of domains in terms of the number of predicates and capabilities. The number of queries are linear in terms of the number of predicates and capabilities (for loop in line 3 of Alg. 1). Note that the for loop in line 5 only contributes to a constant factor in the running time, as only three hypotheses are possible (lines 177-181). Thank you for the question. We will clarify this in the final version.
>
> ```
> SDMA                   | |P| | No of Capabilities
> Warehouse Robot        | 8   | 4
> Driver Agent           | 4   | 2
> First Responder Robot  | 13  | 10
> Elevator Control Agent | 12  | 10
> Cafe Server Robot      | 5   | 4
> ```
>
> Additionally, our approach is highly scalable  and the parameter $\eta$ is used only to learn the correct probabilities in the effects of capabilities.
>
> **Weakness 3)** Thanks a lot for the suggestion. We did give a very high-level overview in lines 352-358. But we understand this might not be sufficient for audiences not familiar with GLIB. In addition to the text, we will add a short summary of GLIB in the revised version using extra pages.

---

> > ### Comment · Reviewer_Sjai · 2023-08-20
> >
> > I thank the authors for their detailed responses and clarifications. After going over the supplementary plots and responses, my queries have been sufficiently addressed (in particular, on the bounds of the learned model from Fig. 1 in the supplementary material). The work tackles an interesting problem and presents some takeaways that may be insightful for the community. I have raised my rating to a "Borderline Accept".

---

> > > ### Author Response · Authors · 2023-08-21
> > >
> > > Thank you for your review. We are glad that the updated plots in the new supplementary material and our response helped in addressing your queries.

---

### Official Review · Reviewer_Jehr · 2023-07-08

**Soundness:** 2 fair
**Presentation:** 3 good
**Contribution:** 2 fair
**Rating:** 5
**Confidence:** 4

**Summary:**

This paper addresses the problem of creating a user-interpretable probabilistic model of the capabilities of a sequential decision-making (SDM) system (through only interacting with the system as a black-box (rather than inspecting its internal structure, e.g., reasoning dynamics).  In particular, PPDDL is proposed to model the SDM system.  An algorithm is proposed that (1) generates queries that seek to determine the location of predicates  either preconditions or effects for capabilties in PPDDL descriptions and (2) data is collected from repeated interactions with the system (likely through simulation, but also possibly through real-world runs) to both determine the validity of PPDDL descriptions and their component probabilities.  Theoretical results seek to establish that the resulting model is both "sound" and "complete" The algorithm is evaluated on five simulation environments, demonstrating convergence to a model with low variational distance from the true SDM.

**Strengths:**

S1) The problem considered has many real-world applications, is indeed understudied, and is relevant for the planning and RL communities at NeuRIPs.

S2) The paper is relatively well-written and easy to follow.

S3) The use of maximal-likelihood estimation for determining probabilities in the PPDDL descriptions seems appropriate as proposed and in-line with other work on learning models of stochastic systems (e.g., model-based reinforcement learning).

S4) I appreciate that the authors considered both theoretical and empirical analysis of their approach.

S5) The empirical results considered a range of benchmarks to aid in evaluating the generalizability of the approach and its advantage over a baseline (GLIB).

**Weaknesses:**

The primary weaknesses of this paper include:

W1) The algorithm proposed appears to perform a linear search through what is ultimately a combinatorial search space of potential PPDDL descriptions.  The for loop on line 3 of Algorithm 1 loops over each condition location and predicate combination independently to determine whether that predicate appears as a precondition or effect for a given condition in some PPDDL description relevant to the system.

However, it isn't clear how the algorithm can discover PPDDL descriptions when two or more predicates appears as a precondition or effect only in AND combination with other predicates.  In that case, looking at and individual <l, p> pair will not provide enough evidence to determine that the predicate p should be in location l for the relevant condition.

For example, say that condition C has a single precondition (and P1 P2) and a single effect (and P3 P4).  Looking at <precondition P1> or <precondition P2> alone will not generate the necessary data to  determine that either are part of a PPDDL description since they are only relevant together and have no measurable outcomes alone.  Similar for <effect P3> and <effect P4>.  While there are only 2|C| locations where a predicate can appear, its appearance occurs within 2^|P| (power set of P) possible combinations of predicates, and it is completely unclear how your algorithm would find all such combinations for even a single capability that requires an search space of exponential combinations of predicates.  Especially using only \eta simulated trajectories of the system.  And without all such combinations, it is unclear how your ultimate model is either sound or complete.

I can imagine there might be some submodular subset of SDM problems where the algorithm converges to the correct set of PPDDLs.  But even in that subset of problems, a certain order of search in the for loop on line 3 (since M* is incrementally constructed) seems highly important for the soundness and correctness of the final model M constructed by the model.

Altogether, the combinatorial nature of the PPDDL descriptions also implies scalability concerns as the number of capabilities and especially predicates increases, but a large number of unique predicates (and hence a very large number of combinations) are likely to be needed for realistic systems of important real-world problems.

W2) The algorithm presented in Algorithm 1 is not any anytime, algorithm, so I'm not sure what it meant to increase the learning time.  It has two for loops whose time complexity depend on the number of capabilities and predicates, and the number of simulation traces \eta was fixed in the experimental setup to 5.

W3) The uncertainty present in the benchmarks seems rather small, so they do not seem to highlight how well the approach handles non-deterministic environments (which is considered one of the main strengths when compared to the prior work in Section 7).  Requiring only \eta = 5 simulation traces per query for good convergence implies that (1) it was rather simple to experience traces that demonstrate the existance (or absence) of a predicate as a precondition or effect (i.e., the environment is not very stochastic),  and (2) the probabilities in the domain must all be close to multiples of 20% (since that's the best level of precision you can achieve with 5 traces in your data-driven Maximal Likelihood Estimation.  In most Monte Carlo sampling of complex stochastic environments, many traces are required (and are only guaranteed to converge as \eta approaches infinity).

W4) I think this work is a good starting point to achieve it's goal of explaining SDM systems to users.  It wasn't clear to me how well the approach factors in the user's environment, which will be necessary for end users (especially non-AI specialists) will interact with and trust the system.  For example, I might receive a set of descriptions of the capabilities of a retail robot vacuum with this approach, but when I take it home to my environment that is different from the one where the PPDDL descriptions were created, the robot might behave very differently.  Maybe the probabilities change as it has a more difficult time navigating around my furniture or along the type of rugs on my floor.  Or maybe even different predicates would need to be added since there are confounding factors (e.g., the presence of different types of pets).

**Questions:**

Q1) How does your search handle the exponential number of possible combinations of predicates that could exist as preconditions or effects?

Q2) In your experiments, what do you mean by learning time, and how did you test your solution with different amounts of learning time?  My naive assumption would be that it took 4 hours to run the entire algorithm and you measured the quality of M* along the way, but that would imply that only looking at a few of the <l, p> pairs (the first few in the for loop on line 3) gave you a really accurate model, which doesn't make sense because they tell you little about the other <l, p> pairs not yet considered.

Q3) How do you interpret the low variational distance with only \eta = 5?

### Post-Rebuttal ###

I thank the authors' for their rebuttal and the ensuing conversation.  They helped strengthen my understanding of the proposed method.  I  think that adding more detail to the body of the paper (instead of the supplement) would greatly strengthen the impact of the work since it appears there are a lot of necessary details that were not originally presented that affect the efficiency and effectiveness of the algorithm.

**Limitations:**

There wasn't a lot of discussion of the limitations of the approach, but addressing many of the weaknesses (especially the combinatorial nature of the problem) would aid the reader in better understanding when the approach could be applicable vs. when improvements to the technique would needed.

---

> ### Author Rebuttal · Authors · 2023-08-10
>
> Thank you for the detailed feedback and questions. We address your questions and other concerns below:
>
> **Q1)** The reviewer correctly points out that the search space of possible preconditions and effects is exponential. We also mentioned this in lines 40-42. Verma et al. [1] showed that preconditions (and effects) that are a conjunction of predicates can be learned in a linear number of queries (in terms of the number of predicates and capabilities) using query synthesis over abstract models (for deterministic settings). Our approach uses the same methodology and performs query synthesis over abstract models in non-deterministic settings.
> This approach leads to fewer queries because the reasoning about the correctness of a precondition or effect is not done using models that are at the same level of abstraction as the ground truth model but instead using a high-level abstract model that has fewer predicates in the precondition and/or effect of some/all capability(ies).
>
> E.g., consider a capability with a precondition $p1 \land p2$ as suggested by the reviewer. The automated query generation process will involve executing the capability successfully in some state $s$ by the policy. The SDMA can only execute the capability in $s$ if $p1 \land p2$ is true in $s$. As mentioned in lines 279-284, if $s$ doesn’t fulfill this criterion (i.e., the SDMA fails to execute the policy successfully) a new query is generated from a new initial state $s’$. Hence, this property of executing the capability in a state having $p1 \land p2$ is ensured. Now, when reasoning about p1, the policy can ask the agent to execute that capability in the state $s \setminus p1$ and if the SDMA fails to execute it then it means $p1$ is part of the precondition. Similarly, this can be done for p2 independently.
>
> In the worst case, the search for a state $s$ where a query policy is executable will be exponential, but as the evaluations show, we can learn the correct model much faster. We also mention a way to overcome this in lines 282-284. Please note that even for methods like reinforcement learning, the worst-case upper bound is exponential in terms of the state space. We will include this discussion in the final version using the additional page permitted by NeurIPS.
>
> **1.1:** About the method working only for submodular PPDDLs: As shown in the results, this method **does** work for PPDDLs that have conjunctive preconditions. E.g., in Fig.2 in the paper, we have the precondition (empty-arm) $\land$ (has-charge) $\land$ (robot-at ?location) $\land$ (at ?location ?item). This capability is from the cafe server robot.
>
> The empirical evaluation showed that QACE (our approach) can learn such models much faster than the closest SOTA approach GLIB as shown in Fig. 5.
>
> **Q2)** Your intuition is correct. For an experiment run, we run QACE as well as the baselines from scratch. For the plots, we took snapshot of the learned models every 60 seconds and computed the variational distance using a fixed test dataset. As you notice in the graph, the variational distance is very high initially, and it drops till the learning process of QACE ends (marked by a blue x on the plots). We do not need to run QACE beyond this point and this time is short for all the domains. On the other hand, GLIB doesn’t have a clear ending criterion. Hence we let it run for 4 hours and see that even with the extra time (and hence extra samples), it cannot learn a better model.
>
> About getting an accurate model with few $\langle l,p \rangle$ pairs: This is not true. Since the plots are shown for a period of 4 hours, it would seem that QACE learns the model without using all $\langle l,p \rangle$ models. We do consider all $\langle l,p \rangle$ models and learn the final model in an efficient manner. Fig. 1 in the extended supplementary material (uploaded with the rebuttal response) clarifies this point using the zoomed-in portions for duration when QACE is running. This figure also shows that the model learned by QACE gets better with time as it processes more $\langle l,p \rangle$ pairs.
>
> This also addresses a similar concern raised in W2. Please note that a larger  $\eta$ might be needed for a more complex domain to learn the correct probabilities.
>
> **Q3)** The hyperparameter $\eta$ is 5, but this does not mean that we execute that capability just 5 times. As mentioned in the paper, consider $\mathcal{P}$ to be the set of predicates, and $|\mathcal{P}|$ be the number of predicates. Now a capability $c$ can appear in policies for $\langle l, p \rangle$ pairs, such that location $l$ corresponds to a precondition or effect in $c$. So effectively, $c$ can appear in at least for $2 \times |\mathcal{P}|$ queries. So we will have at least $2 \times |\mathcal{P}| * \eta$ samples for each capability.
>
> **W4)** This is precisely the motivation for our work, and is addressed directly by the presented method. In your scenario, the user would be able to use our system to discover the new model of the agent, with probabilities in this new environment. We agree that predicate discovery is also an open problem in this area and we plan to address it in future work by building upon the presented methods.
>
> ----------------------
> *References:*
>
> [1] Verma, P., Marpally, S. R., & Srivastava, S. Asking the Right Questions: Learning Interpretable Action Models Through Query Answering. AAAI 2021.

---

> > ### Comment · Reviewer_Jehr · 2023-08-18
> > **RE: Rebuttal by Authors**
> >
> > I thank the authors for their responses to my questions and overall review!  I especially better understand your answer to my Q2, and relatedly I noticed that I missed the X in Figure 5 that indicates when QACE stopped running.
> >
> > I'm still confused about Q1 -- where in your Algorithm 1 will you consider p1 AND p2 together?  Line 3 is a loop over all <l, p> pairs, and p is an element of the set of possible predicates P, so wouldn't p be only a singleton and not a conjunction of elements of P?
> >
> > And if discovering p1 AND p2 is dependent on testing in a particular state s where p1 AND p2 are required, how do you insure that QACE receives that state s as an input?  Do you run QACE on every possible state?

---

> > > ### Author Response · Authors · 2023-08-19
> > >
> > > Thank you for going through our response.
> > >
> > > The method does work iteratively, but the iterations are not independent of each other. Consider the case where QACE processes $\langle l,p \rangle$ = $\langle$ precondition of $c, p_1 \rangle$ in line 3. This iteration of the loop will involve creating a query with an initial state where $p_1 \land p_2$ is true (more on how we get this state later). Using this query, QACE will set $M^*$ to have $p_1$ as a precondition of capability $c$ in line 8. Next, when QACE considers, $\langle l,p \rangle$ = $\langle$ precondition of $c, p_2 \rangle$, the three hypotheses generated in line 4 using $M^*$ will have the precondition of $c$ as $p_1 \land p_2$ (in $h_T$), $p_1 \land \neg p_2$ (in $h_F$), and $p_1$ (in $h_I$). So essentially, QACE builds upon the already learned partial models in previous iterations.
> > >
> > > **About getting the states where a capability is executable:** This refers to getting the state where $p_1 \land p_2$ is true in the example above. Your intuition is correct that, in the worst case, QACE will have to check all possible states that can be generated using the predicates. In practice, though, we use directed exploration (also mentioned in response to reviewer *k56H*) to avoid such cases. We start with the input state $s$ (we only use one state as input) and use the partially learned model $M^*$ to generate new states where $c$ might be executable. Empirically, this process is very fast, as evident from the results. If this approach does not work, it defaults to a randomized exploration to generate a state where $c$ is executable, which can lead to exploring all states in the worst case.

---

> > > > ### Comment · Reviewer_Jehr · 2023-08-21
> > > > **Official Comment by Authors**
> > > >
> > > > I thank the authors again for their response.  In your example, if p1 is only part of the precondition of c in conjuction with p2, then it is still unclear to me how looking at a state where p1 and p2 is true will result in p1 being added as a precondition of c in M*.
> > > >
> > > > Maybe it will help to look at a specific example?  Let's say the task is a robot preparing tea for a user.  The capability c in question is the capability to hand over a hot cup of tea to the user (necessary for the final step of the task).  The success of this capability has the precondition that p1 = the robot is holding a cup and p2 = the cup contains hot tea.  If the algorithm first considers the capability "c = handing over tea" and the only precondition considered thus far is "p1 = the robot is holding a cup", how do we know that this precondition will be added to the model?  Without considering simultaneously "p2 = the cup contains hot tea", we cannot know whether p1 is part of a precondition based on the outcome of the environment, correct?
> > > >
> > > > There seems to be an assumption that the algorithm will have state s where the robot is indeed holding the cup and the cup contains tea.  But why is that the state it considers and not the many other states where the robot might be holding a cup but the cup doesn't have hot tea (the cup could be empty, it could contain cold water, it could contain hot water, it could contain only the tea bag, etc.).  If any of those other states were considered, then I'm not sure why p1 would be noticed as a precondition of c.
> > > >
> > > > I'm sure I'm missing an important step, so you could be please elaborate?

---

> > > > > ### Author Response · Authors · 2023-08-21
> > > > >
> > > > > Thank you for taking the time to review this work and engage in active discussion. We understand the confusion.
> > > > >
> > > > > **Initial state $\mathbf{s}$**: In your example, your understanding is correct that to learn the precondition correctly, we need the agent to execute the capability in any state where the robot is indeed holding the cup and the cup contains tea (let us call such set of states $S’$. Here $S’$ is a set of states because many states can have this property where some other object is in a different location or position, etc., but in all of them, the robot is holding the cup containing tea). But, please note that the *input state $s$ need not be in $S’$*. $s$ can be anything. For this discussion, consider $s$ where the robot is holding an empty cup.
> > > > >
> > > > > QACE just needs one instance where the agent successfully executes the capability $c$, which can be done in 2 ways.
> > > > >
> > > > > 1. The policy generation process can take the input state $s$, and the policy might be to execute the capability “fill the cup with tea” (until it succeeds in case the action can fail with some probability), and then in the state where the robot is holding the cup with tea, the policy will be to execute the capability “handing over tea.” Figure 1 in the main supplementary material shows an example of such policies. We also show in proposition 1 (main supplementary material) that since we are considering the capability of “handing over tea” in a particular iteration, the query policy generated in that iteration by QACE will involve executing this capability.
> > > > >
> > > > > 2. If the query generation succeeds, but it involves the agent executing the capability in a state where it cannot execute it. As a fail-safe measure, QACE can perform a directed exploration to find a state in $S’$. It can be any search technique like BFS, A*, etc. Using this exploration over the complete state space $S$, QACE can generate new states to replace $s$ in the query generation process. Note that this exploration to search for new initial states might not be needed at all.
> > > > >
> > > > > So the main property here is that QACE can search for the state where the agent can successfully execute the capability $c$.
> > > > >
> > > > > **How QACE learns a precondition from such a state?**: Consider that QACE is considering the pair $\langle h_T, h_F \rangle$ in line 5. In the example being considered, executing $c$ in a state $s' \in S'$ can help the agent distinguish between $h_F$ and $h_T$. Here the model corresponding to $h_F$ will be unable to execute $c$ in $s'$, whereas the model corresponding to $h_T$ and the agent will be able to. So QACE will prune $h_F$ in line 8. Next, it will consider $\langle h_T, h_I \rangle$. Here, to distinguish between the models corresponding to these hypotheses, the agent will execute $c$ in a state where the robot is not holding the cup and the cup contains tea (equivalent to $\neg p_1 \land p_2$). Note that this state is also not generated manually, and the query generation does this autonomously, starting from the same state $s$. Here the model corresponding to $h_T$ will fail to execute the capability $c,$ $h_I$ will succeed, and the agent will also fail. Hence QACE can prune out $h_I$, leading it to learn the correct hypothesis $h_T$ that $p_1$ is a precondition of $c$.
> > > > >
> > > > > We will use an example like this in the supplementary material to clarify the process in the final version.

---

> > > > > > ### Comment · Reviewer_Jehr · 2023-08-22
> > > > > > **RE: Offical Comment by Authors**
> > > > > >
> > > > > > I thank the authors again for their clarification!  I think I have a better understanding of how the process works.  I was focused pretty heavily on just the content of Algorithm 1 since it is the only algorithm outlined in the paper, but it sounds like there are several key processes that work in combination with Algorithm 1 to support its efficiency and effectiveness.  Rather than add these important details to the supplementary material, I would strongly suggest adding them to the main body of the paper so that the discussion is more complete, the work is more reproducible, and readers are more likely to follow what is happening so they can employ the technique or build upon it.

---

### Official Review · Reviewer_pN4U · 2023-07-22

**Soundness:** 3 good
**Presentation:** 3 good
**Contribution:** 3 good
**Rating:** 5
**Confidence:** 4

**Summary:**

This paper proposes a method for modeling the capabilities of black-box artificial intelligence systems, which can plan, act, and execute in a stochastic setting. Specifically, the proposed method introduces an active learning approach to interact with the black-box SDM system and learn a probabilistic model that describes its functionality. The paper presents theoretical analysis that guarantees the convergence of the learning process. Empirical evaluations on different intelligent agents and simulated scenarios demonstrate that the proposed method exhibits generalizability with limited data and effectively describes the capabilities of the agents.

**Strengths:**

1. The paper proposes a method for modeling the capabilities of black-box artificial intelligence systems. Overall, this paper is well-motivated and provides detailed explanations.
2. This paper describes how the active learning method effectively interacts with the black-box SDM system and introduces a probabilistic model that explains its functionality.


**Weaknesses:**

1. The empirical evaluation results of the paper demonstrate that the model can adapt well to training tasks with few samples. However, how does it perform in terms of generalization evaluation for new tasks?
2. Another major concern I have is the generality of the proposed method, especially when people want to apply it to more complex manipulation tasks. While it has been validated on several examples, I am unsure how these ideas can be extended. How can it be applied to more practical environments with skill-specific parameters, such as grasping angles and placement targets?
3. In practice, how is the intelligent system trained with data? How is this data collected on physical robots, and what are the challenges involved?


**Questions:**

Regarding this paper, please refer to my "Weaknesses" for questions and comments. They mainly concern the limitations of the method, evaluation, and practical data.

**Limitations:**

The paper briefly mentions its limitations, but it would be beneficial to include a discussion on potential social implications.

---

> ### Author Rebuttal · Authors · 2023-08-10
>
> We thank the reviewer for questions and support. We address your comments on the weakness of the approach below:
>
> **Weakness 1)** The test set is not the same as the training set. As mentioned in lines 332-334, we used a single problem as input. Additionally, QACE (our approach) generates queries and based on their responses we learn the correct model. The methods were tested on environments that were much larger in terms of objects as compared to the input problem. This shows the generalization of our approach for tasks that follow the same dynamics.
>
> **Weakness 2)** Our approach is highly general for SDM agents. Our experiments on the SDMA with cafe server robot had complex grasping poses and angles. The predicates used to express the domain were at a high level of abstraction which abstracted this low level information and hence explains the high-level dynamics of the system. For vocabularies that can express the difficult concepts like different types of grasps, this approach will accommodate for those predicates and learn a model in terms of those predicates. This feature makes the models personalized for each kind of user.
>
> **Weakness 3)** Our intelligent data gathering process uses active learning based query generation allowing the collection of training data in an automatic, handsfree fashion without requiring any human intervention. In our paper, this process was accomplished on benchmark SDM agents connected with simulators.
>
> In the real-world, on physical robots, our system could connect to the robot via an interface or be deployed directly. It can issue commands (execute capability c) and observe the responses to those commands for collecting training data. The key challenges here would pertain to safe operation of the robot. Since the models of its capabilities are not known, running these commands directly on the robot could potentially lead it to perform an unsafe operation. This could be easily circumvented if the physical robot was inbuilt with rudimentary safety protocols. This is part of the future extension of the work that we mention in lines 422-424.

---

> > ### Comment · Reviewer_pN4U · 2023-08-16
> >
> > I thank the authors for their rebuttal and their work.
> >
> > I read the rebuttal carefully. I've raised my rating to weak accept because some of my concerns have been addressed in the rebuttal, and the work offers some interesting ideas that should be shared with the community.

---

> > > ### Author Response · Authors · 2023-08-21
> > >
> > > Thank you for your review. We are happy to see that we have addressed your concerns. It would be great if you could update the score to reflect your comments. Thank you.

---

### Author Rebuttal · Authors · 2023-08-10

We thank the reviewers for their detailed reviews and comments. We answer the questions posed by the reviewers separately. Please find them in the response below the reviews. We are also adding a supplementary page with two plots. One showing the zoomed in version of the plot for variational distance vs learning time for QACE (our approach) and GLIB (baseline). And the other one showing the variational distance vs number of queries for QACE.

---

### Decision · Program_Chairs · 2023-09-21

**Decision:**

Accept (poster)

**Comment:**

I am recommending accept for this paper. The reviews for the paper were originally (6, 5, 5, 5, 4). The reviewers praised the paper for being interesting and novel, saying it was a “new direction to reason about the capabilities of block-box SDMA in an interpretable manner”, and a “novel combination of techniques”, and said that "the results are important”. They liked that the technique had real world applications, and praised the theoretical analysis (they liked the completeness and soundness results) and empirical analysis (for being conducted in a range of environments).

The main complaints about the paper were that the method may be too complex, concerns were raised about the computational complexity and scalability, and about who would actually use it, and whether it would be possible to compare to simpler baselines. In my opinion the authors adequately addressed each of these concerns in the rebuttal. For example, regarding the simpler baselines, they explained that the baseline they implemented (GLIB) is essentially what the reviewer was requesting, and that “to the best of our knowledge, no other existing approach is able to solve the problem we are addressing in this work”.

The one negative reviewer (reviewer Jehr, who originally assigned a score of 4) raised concerns about how the method can be applied with conjunctive predicates. The authors attempted to answer his concerns to explain how this could work, and pointed out that one of their examples in Figure 2 includes conjunctive predicates. However the reviewer was still not clear on how the method worked, and asked for further clarification without raising their score. During the discussion with reviewers, I asked reviewer Jehr whether they believe the paper should be accepted if the method were explained more clearly, or whether they would argue for rejecting the paper. They responded that they would not argue for rejection, but that the method should still be clarified further, but ended up raising their score to a 5.

Therefore, I am recommending accept, although given the borderline scores (average of 5.2), I'm recommending it is accepted as a poster. I strongly encourage the authors to take the opportunity to revise the manuscript in the camera-ready version in order to address reviewer Jehr's concerns, particularly to clarify how the method works. Reviewer Jehr also requested that details necessary for understanding the efficiency and effectiveness be included. It is not only better for the scientific community, but in the authors' best interest to clarify the manuscript, because if the paper is more clear and comprehensible it is more likely to be cited and built upon in future work.